# An Experimental Electronic Board ADF339 for Analog and FPGA-Based Digital Filtration of Measurement Signals

Cezary Pałczyński [†] and Paweł Olejnik *,[†] 

Department of Automation, Biomechanics and Mechatronics, Lodz University of Technology, 1/15 Stefanowski Street, 90-537 Lodz, Poland; 228446@edu.p.lodz.pl
* Correspondence: pawel.olejnik@p.lodz.pl
[†] These authors contributed equally to this work.

**Abstract:** This work introduces and examines a new programmable electronic system, Board ADF339, designed for filtering analog measurement signals of low frequencies. The system operates in a mixed mode in collaboration with a digital controller implemented on the myRIO-1900 FPGA module. It enables the digital selection of the type and frequency settings of the UAF42 integrated circuit. In the technical implementation section, electronic filter and phase shifter circuit diagrams are presented, along with the digital counterpart of the analog filter. Tests of this system were conducted on signals generated using a function generator, which was followed by the filtration of signals occurring in real laboratory setups. A series of real responses from three different laboratory systems and a measurement system utilizing LabVIEW FPGA virtual instruments are demonstrated. After computing SNR indicators for noisy waveforms, the application scope and usability of the board are highlighted.

**Keywords:** electronic system design; analog filter; digital filter; programming FPGA; myRIO-1900





## 1. Introduction

Digital signal processing, known as DSP, has grown significantly with applications in various fields. Filters play a crucial role in signal processing, and digital filters, preferred over analog ones, offer an optimal frequency response, high reliability, and multi-rate processing capabilities. Digital filters are categorized into finite impulse response (FIR) and infinite impulse response (IIR). FIR filters, which are stable and have advantageous properties, are often favored due to their linear phase design potential. While the current literature in this engineering field is extensive, we highlight the significance and application of a specific component used in the proposed mixed filtering mode experimental electronic board.

### 1.1. Field Programmable Gate Arrays as Digital Filtering Units

An efficient high-pass filter implementation using an arithmetic optimization algorithm, demonstrating superior results compared to the Park–McClellan algorithm in terms of pass-band ripple and stop-band attenuation, was introduced in [1]. Additionally, the article discusses the hardware implementation of the proposed design on the FPGA platform.

The article [2] presents the design of a 5th-order tunable IIR digital low-pass filter using MATLAB with the FDATool toolbox. The filter operates from 20 to 70 MHz at a 176 MHz sampling frequency, with 14 input bits and 15 output bits. Characteristics such as frequency response, group delay, stability, pole and zero locations, and system noise are considered. The use of approximated filter coefficients enhances circuit simplicity and increases the sampling frequency, thus resulting in a maximum frequency of 175.596 MHz on an FPGA, as confirmed by systematic simulation results.

Another FPGA-based digital filter was designed in [3] to remove noise from ECG signals. The study proposes an efficient ECG denoising technique using FPGA. The authors

employed high-pass, moving average, and Savitzky–Golay filters, which they tested on diverse MIT-BIH ECG signals with added noise. Performance metrics like SNR, MSE, and COR were used for evaluation. The system was implemented in VHDL and simulated on Vivado.

### 1.2. Industrial Applications

Industries widely employ filtering methods across various domains such as telecommunications, biomedical, signal processing, and testing. Optimal techniques, ensuring high speed and minimal distortion, are crucial in both analog and digital applications [4–6].

The work of [7] delivers a study on the removal of unwanted noise from a pulse wave in high-energy medical linear accelerators designed for cancer treatment. The authors used NI Multisim, MATLAB, and LabVIEW to generate and test the pulse wave prototype. They developed VHDL code for a low-pass digital FIR filter using the FDATool, with a response similar to a Bessel filter. The filter was implemented on a Spartan 3A XC3S700A series FPGA and simulated using Xilinx ISE software. The study concluded that FPGAs offer advantages in digital filtering, including cost-effectiveness, design flexibility, high performance, and low power consumption in real-time (RT) implementations.

### 1.3. NI myRIO-1900 in Practical Filter Designs

The paper [8] discusses the implementation of a maximally flat FIR differentiator with cost-effective parameters. By appropriately selecting these parameters, the filter coefficients exhibit a common divisor reciprocal to a power of two. This characteristic enables an efficient implementation using shift and add operations, which are particularly advantageous for FPGA implementation. The authors conducted tests on National Instruments myRIO-1900, a hardware system featuring analog and digital inputs/outputs, a dual-core ARM Cortex A-9 processor, and an integrated FPGA. The proposed differentiator was successfully implemented on an FPGA and tested in real time. The article concludes with a comparison of FPGA resource utilization between the proposed maximally flat FIR differentiator and a differentiator designed through a common approach.

Passive LLCL filters are commonly used in power electronics to improve the performance and reliability of power supply systems by filtering out unwanted harmonic content and noise. The filter is named after the order in which the components like inductors and capacitors are arranged in the circuit. The work of [9] addresses the growing issue of power quality deterioration caused by harmonics generated by power electronics devices. It introduces a solution using the LLCL filter and a harmonic mitigation monitoring RT system employing fast Fourier transform and an NI myRIO-1900 device. Testing with a nonlinear load demonstrated the effectiveness of the approach, thus reducing the total harmonic distortion when the filter was applied.

The study [10] focuses on acquiring signals generated by the electrical activity of the heart using an NI myRIO-1900 embedded device. Due to the low current manifestation of heartbeats, signal amplification was necessary for processing. The project employed a two-stage amplification process utilizing an integrated AD620 for pre-amplification and an OA TL084 for post-amplification. Real-time data acquisition took place in the FPGA device. To prevent data loss due to varying operating frequencies between the processor and the FPGA, processor FIFOs were employed. The acquired data were processed using wavelets to separate and display distinct components of the ECG.

### 1.4. Application of the UAF42 Integrated Circuit

Over the past decade, programmable active filters, particularly those employing switched capacitor topologies, have gained popularity. These filters allow for the easy adjustment of parameters like natural frequency $\omega_n$ and quality factor $Q$ by varying the clock frequency. However, switched capacitor filters are susceptible to issues such as clock feedthrough noise and aliasing errors due to their sampled data nature. Burr-Brown offers the UAF41 and UAF42 as ICs for universal filters, thereby addressing these concerns. The

UAF42, in particular, facilitates the construction of an analog, digitally programmable filter with two poles. This monolithic, state variable-active filter chip minimizes sensitivity to external component variations and effectively eliminates aliasing errors and clock feedthrough noise that is common in switched capacitor filters. Additionally, it provides simultaneous low-pass, high-pass, band-pass, and band reject (notch) outputs. The most popular and useful circuits that realize such functions are presented in [11] together with their design equations.

The signal from the motion sensor developed in [12] underwent filtering and amplification using a feedback control circuit based on UAF42. This circuit, with finite bandwidth, realizes the system's transfer function. A high-pass UAF filter eliminates noise, and adjustable resistor and capacitor values tune bandwidth and quality. The filtered signal is then amplified and sent to a voltage-controlled current source, thus governing the voice coil's current proportionally to the input voltage.

The article [13] discusses a novel system in biomedical engineering for acquiring and processing human cardiovascular electrical signals. The system utilizes a virtual instrument concept incorporating an INA128 weak signal amplifier, a universal active filter chip UAF42, and a PCI6024E data acquiring board to sample data into a computer. The software, developed on the LabVIEW platform, employs a difference threshold algorithm to detect R waves and calculate R-R duration. The system was tested on standard MIT/BIH ECG signal data and applied in real-time analysis of ECG signals from patients.

The challenges of surface electromyography (SEMG) due to its susceptibility to external noise and weak characteristics are discussed in [14]. It emphasizes the importance of establishing a suitable acquisition and amplification system. The INA128 chip served as the core IC for the SEMG amplifier, and the UAF42 IC functioned as a notch filter to eliminate the 50 Hz frequency. The feasibility of the SEMG amplifier was simulated and tested, thereby demonstrating its effectiveness in helping amputees acquire electromyographic signals.

*1.5. Originality of This Work in the Perspective of the Literature Review*

A few applications of analog and digital filters have been discussed. These include filters implemented using FPGA, their industrial applications, the NI myRIO-1900 embedded system, and the UAF42 integrated circuit, all of which were utilized in this study.

A significant original aspect explored in this work is the joint operation between an analog filter constructed based on the UAF42 circuit and a digital filter built on the NI myRIO-1900 embedded system (mixed mode). Moreover, this joint operation enhances the tuning of the parameters, frequency response, phase shift, and consequently, the range of applications. This is achieved, for example, through digital adjustments of the quality factor and cutoff (central) frequency or correction of phase shift. The elaborated ADF339 board was validated through experimental measurements on three original laboratory test stands, thus leading to enhancements in its performance.

A low-pass filter was designed in [15] with electronically adjustable characteristics, thus allowing for reconfiguration of the order (first to forth order functions available) and electronic control of the output response's cutoff frequency. The electronic adjustment of the approximation characteristics was explored for Butterworth, Bessel, Elliptic, Chebyshev, and Inverse Chebyshev approximations. The design was validated through PSpice simulations and experimental measurements.

From the other side, a tunable mixed mode analog filter was proposed in [16]. The designed circuit has all five filter functions—low-pass, high-pass, band-pass, band reject, and all pass—by selecting appropriate input signals. The parameters of the proposed circuit, namely cutoff frequency and quality factor, are independent, with the frequency being electronically tunable.

Anti-aliasing analog filters like the RC (low-pass) filter are positioned before ADCs. An ideal anti-aliasing filter exhibits unity gain in the pass-band with no gain fluctuations and

provides alias attenuation matching the theoretical dynamic range of the data conversion system; see [17].

The diverse range of applications discussed underscores the indispensable role of analog filtering considered in this work in demanding real-world implementations.

*1.6. Organization of This Work*

After reviewing the current applications in Section 1, a mathematical background developed for the practical realization of certain filter configurations has been introduced in Section 2. Section 3 provides a description of the electronic board, while the subsequent Sections 4 and 5 present useful diagram coding for the FPGA and the results of filtering various real measurement signals with different forms of the noise component.

Finally, the study and experiments are concluded, thusproviding practical directions for its use and suggesting avenues for further development.

## 2. Mathematical Background

Network functions associated with linear time-invariant lumped circuits can be expressed as rational functions of $s$ representing the quotient of polynomials in $s$. The order of the circuit is determined by the degree of the polynomial of $s$ present in the denominator of the transfer function. The most general form for a second-order rational function in $s$ is the biquadratic form:

$$F(s) = \frac{A_z s^2 + B_z s + C_z}{A_p s^2 + B_p s + C_p} = K\frac{(s - z_1)(s - z_2)}{(s - p_1)(s - p_2)} \tag{1}$$

Here, $z_i$ represents the zeros and $p_i$ represents the poles of the function $F(s)$. The constant $K = A_z/A_p$ is independent of frequency.

The general form of a second-order biquadratic function is represented as follows:

$$F(s) = K\frac{s^2 + \frac{\omega_{oz}}{Q_z}s + \omega_{oz}^2}{s^2 + \frac{\omega_{op}}{Q_p}s + \omega_{op}^2} \tag{2}$$

Parameters $Q_z$ and $Q_p$ denote the quality factors associated with the zeros and poles, respectively, and $K = \frac{A_z}{A_p}$. The frequencies $\omega_{op}$ and $\omega_{oz}$ represent the natural frequencies of the system described by $F(s)$ or the frequencies of its poles and zeros, respectively. The expressions for the quality factors and frequencies of the poles and zeros in terms of the polynomial coefficients are given by the following:

$$\omega_{oz} = \sqrt{\frac{C_z}{A_z}}, \quad Q_z = \frac{\omega_{oz}}{\frac{B_z}{A_z}}, \quad \omega_{op} = \sqrt{\frac{C_p}{A_p}}, \quad Q_p = \frac{\omega_{op}}{\frac{B_p}{A_p}} \tag{3}$$

Poles and zeros generally exist as complex conjugate pairs. Real poles or zeros are characterized by having zero imaginary parts. The poles of the function $F(s)$ can be expressed as follows:

$$s_{1,2} = -\frac{B_p}{2A_p} \pm \sqrt{\frac{B_p^2}{4A_p^2} - \frac{C_p}{A_p}} = \omega_{op}\left(-\frac{1}{2Q_p} \pm \sqrt{\left(\frac{1}{2Q_p}\right)^2 - 1}\right) \tag{4}$$

In cases where $Q_p > 0.5$, the poles exhibit a complex nature.

Filter types designed and implemented on the experimental board ADF339 do not require the inclusion of the second term at $s$ in the numerator of Equation (2). Therefore, the quality factor $Q_z$ is not considered, and symbol $Q$ will be replaced with $Q_p$. The parameters and the transfer functions of the analog filter circuit are given in Sections 3.2 and 3.3.

In analog filter design, the focus is on circuits that feature easily adjustable quality factors and natural pole frequencies. An additional and advantageous attribute in practical circuits involves the capability to independently adjust the quality factor apart from the corresponding pole frequency.

The introduced mathematical description of basic filter models will be used with respect to the setting of parameters of the universal analog filter board ADF339.

## 3. Design of the Experimental Electronic Board ADF339

To compare filtration methods and assess them, a measurement system based on the new experimental electronic board ADF339 shown in Figure 1 was constructed by incorporating both analog and digital filtering. The control of the parameters and observation of real signals were implemented using a PC as an HMI panel; the abbreviations are as follows: AWG—arbitrary waveform generator; CPU—host computer with program in LabVIEW. The block diagram of the measurement system shows in a simplified form the communication between the subsystems of the entire laboratory setup.

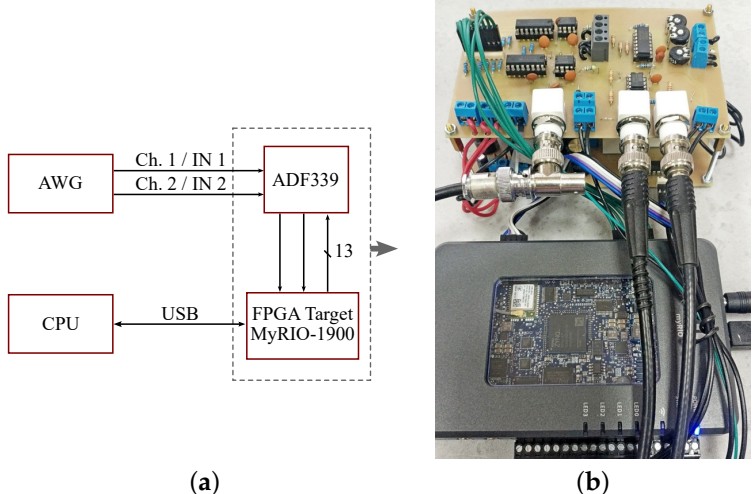

(**a**)  (**b**)

**Figure 1.** Block diagram of the experimental board (**a**) and picture of the data acquisition system (**b**) comprising two stacked circuit boards (logic above and power below) and myRIO-1900 device with Xilinx Zynq-7010 processor (LabVIEW RT) and FPGA.

The complete electrical schematic of the expanded analog filter circuit ADF339 is presented in Figures 2–5. The programmable analog filter circuit was designed in two parts due to the concept of constructing the system with two PCB boards connected by wires. The power supply circuit consists of a transformer, rectifier diodes, capacitors, and regulators to achieve symmetrical ±9 and +5 V power. The electronics design follows the well-known standard scheme, e.g., that given in [18].

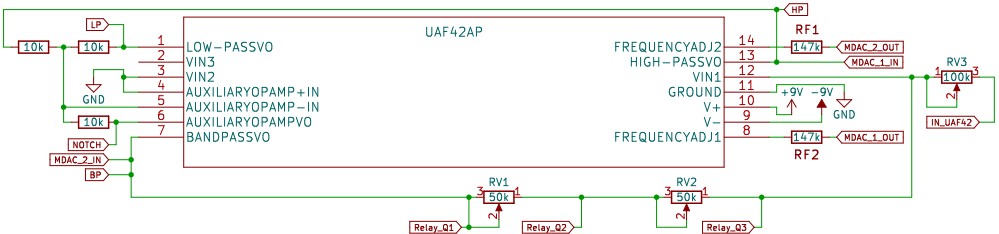

**Figure 2.** The main unit including the digitally programmable analog UAF42 state variable filter.

The design is based on the universal active analog filter circuit UAF42P TI. The circuit features a single input labeled 'IN UAF42' for the analog signal and four filter outputs labeled LP (low-pass), HP (high-pass), BP (band-pass), and notch (band reject). The system

is powered by a symmetrical voltage 9 V. The potentiometer $R_{V3}$ sets the resistance to 49.9 kΩ.

The potentiometer $R_{V3}$ additionally allows for adjusting the gain within the pass-band of the filter. For low-pass, high-pass, and notch filters, the formula for gain is as follows: $A = -20 \cdot \log_{10}\left(\frac{R_{V3}}{50k\Omega}\right)$ dB. The value of 50 kΩ is obtained from the internal components of the UAF42 for an $R_{V3}$ setting of 49.9 kΩ. For the mentioned filters, $A = 0.0174$ dB, meaning the setting of $R_{V3}$ could as well be 50 kΩ. In the case of a band-pass filter, to calculate the gain within the pass-band, the formula provided in Section 3.3 is used.

Potentiometers $R_{V1}$ and $R_{V2}$ are employed to select the filter characteristics (Bessel $Q = 0.58$, Butterworth $Q = 0.707$, Chebyshev with 3 dB ripple $Q = 1.3$). Setting potentiometers to values of 29 and 35 kΩ allows for the selection of Bessel or Butterworth filters by connecting 'Relay Q1' to 'Relay Q2' or 'Relay Q2' to 'Relay Q3' using the relays presented in another part of the circuit.

With both junctions open, the equivalent resistance is 64 kΩ, thus yielding $Q = 1.28$ and closely resembling a Chebyshev filter with 3 dB ripple ($Q = 1.3$). The maximum cutoff frequency for low-pass and high-pass filters (center frequency for band-pass filters) is approximately 1083 Hz. Therefore, based on the resistance, it should be 147 kΩ for two resistors connected to inputs 8 and 14 in UAF42.

To create a notch filter, the operational amplifier within the UAF42P circuit is used, thereby summing the outputs of the low-pass and high-pass filters. The input and output for the first digitally controlled attenuation circuit is represented by 'MDAC 1 IN' and 'MDAC 1 OUT', as found in the datasheet of the AD7541A, and for the second attenuation circuit, the input and output are 'MDAC 2 IN' and 'MDAC 2 OUT', respectively. The digitally controlled attenuation circuits enable the adjustment of the cutoff frequency.

The electrical diagram in Figure 3 depicts two almost identical digitally controlled damping systems. On the ADF339 board, two attenuation circuits were utilized, as the UAF42 integrated circuit requires two attenuation sub-circuits connected to each of the pins labeled 'Frequency Adj1' and 'Frequency Adj2'. Employing two attenuation circuits enables control of the cutoff frequency for low-pass and high-pass filters, as well as the center frequency for band-pass filters implemented by the UAF42. Such a single system consists of an operational amplifier and an 8-bit multiplying digital-to-analog converter (DAC). The DAC setup includes WR and CS inputs connected to GND so that the states on the 8-bit data bus directly control the operational amplifier's gain. In the case of the filter, the DAC settings will alter the cutoff frequency according to the formula $f_g = (x/255) \cdot 1083$ Hz, where $x$ represents the input on the 8-bit data bus.

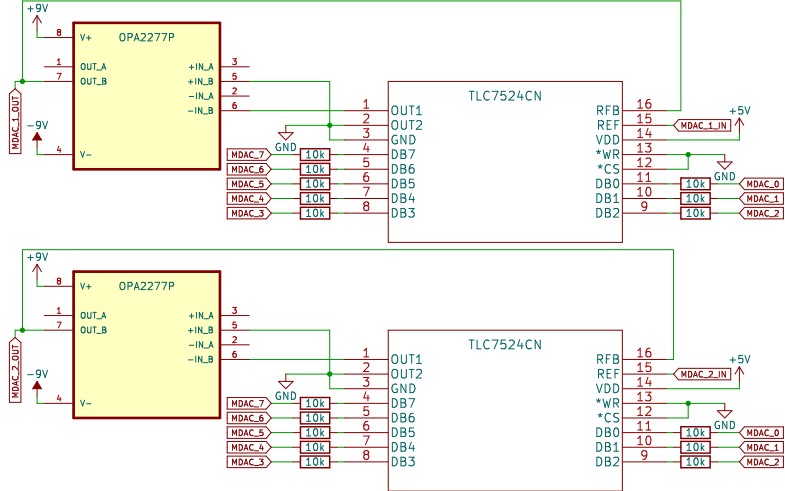

**Figure 3.** Two systems of digitally controlled damping connected to each of the pins labeled 'Frequency Adj1' and 'Frequency Adj2'.

The circuit in Figure 4 depicts a summing of signals from two analog inputs: 'IN BNC 1' and 'IN BNC 2'. Subsequently, this signal is inverted using the second channel of the OPA2277P amplifier.

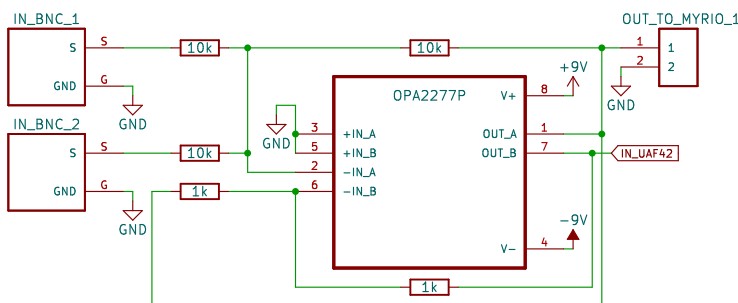

**Figure 4.** Summation unit of two analog signals 'IN BNC 1' and 'IN BNC 2'.

The circuit diagram shown in Figure 5 illustrates various connectors, including power supply terminals ±9 and +5 V, as well as GND, sockets to the myRIO system, and the ULN2803 Darlington driver circuit for controlling relay coils. The power supply unit, implemented using a transformer and voltage regulators (L7805CV, L7809CV, and L7909CV), is not depicted due to its standard implementation, which has been omitted in this work. Additionally, the schematic incorporates voltage filtering capacitors strategically placed near operational amplifier circuits MADC and UAF42P.

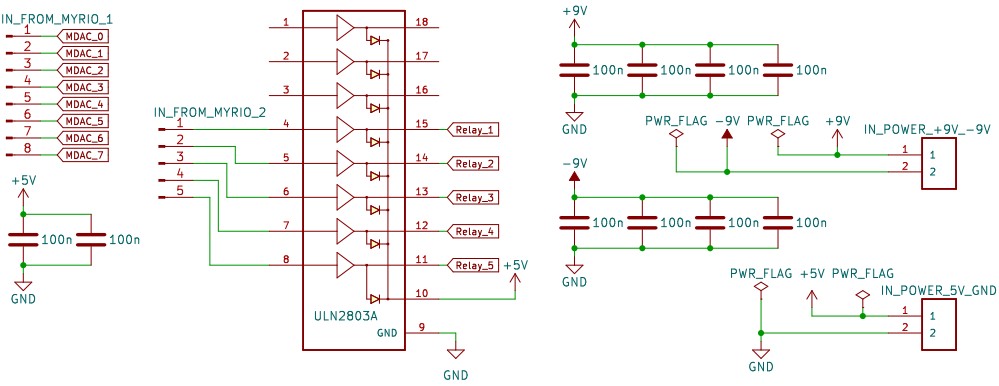

**Figure 5.** Connector to power supply unit (±9 and +5 V, GND) and sockets to the myRIO-1900 and ULN2803 Darlington driver.

### 3.1. Applied Modifications of the TI Model Circuit

The first modification introduced to the model of programmable analog filter circuit proposed by TI involved replacing the no longer available digital-to-analog converters DAC7541A with devices of similar characteristics, namely TLC7524. Operational amplifiers were substituted with the OPA2277PA model. In the model of a programmable filter, four outputs for specific filter types were envisaged: low-pass, high-pass, band-pass, and band stop. To eliminate switching using a switch, an output selector was implemented using three relays; see Figure 6 (left-hand side).

The benefits of using relays instead of ordinary switches include the following: (a) the ability to select the filter output from the LabVIEW program; (b) the possibility of programming in such a way as to enable automatic data acquisition responses for each type of implemented filter; (c) when comparing analog and digital filters, using relays allows for the simultaneous selection of both an analog and a digital filter from a single variable in the LabVIEW program, which is faster and eliminates the potential error of selecting a different type of analog filter than digital and comparing responses.

To reduce electromechanical interference generated by the relay, the relay is powered from a stabilized voltage along with capacitor filters. During filter response acquisition,

data are not recorded during relay switching states, as the dynamic state could generate greater interference.

Proper control of the relays allows for the selection of one input and the transmission of the signal to the output. The next modification is the ability to change the $Q$ factor, which in the UAF42 system is achieved by adjusting the resistance. The use of two adjustable potentiometers as trimmable resistors and two relays allows for obtaining four equivalent resistances; see Figure 6 (right-hand side).

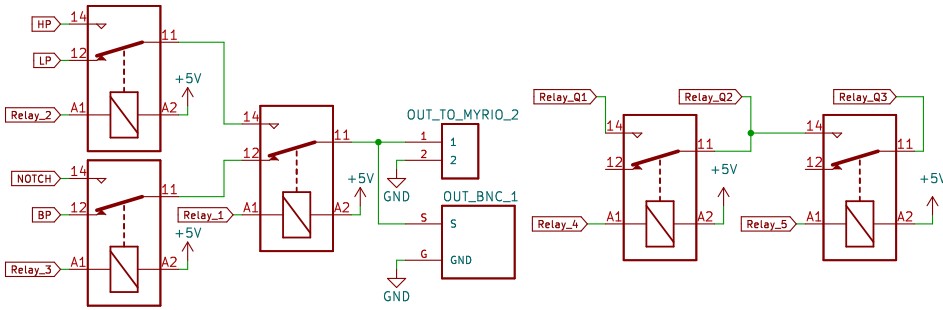

**Figure 6.** Implementation of the filtering output selector using three relays.

The summing amplifier has been extended in the system by adding a signal summing amplifier for two generator channels. Since the UAF42 inverts the phase, an inverting amplifier has also been applied at the output of the summing system.

The digital outputs of the myRIO-1900 device have insufficient current capacity, so to control relays 1–5, a Darlington array, ULN2803A, was used for amplification.

### 3.2. Parameters of the Analog Filter Circuit

The programmable analog filter circuit includes an 8-bit digital-to-analog converter; therefore, the relationship of the maximum cutoff frequency dependent on the $R_F$ resistance (see the digitally programmable analog filter UAF42) is as follows:

$$f_{c,max} = \frac{1}{2\pi \cdot 10^{-9} \cdot R_F} \tag{5}$$

where, according to the scheme in Figure 2, $R_{F1} = R_{F2} = R_F$.

Taking into account that $R_F = 147$ kΩ, the value of the cutoff frequency $f_c$ of the programmable analog filter can be adjusted in the range from 0 to 1083 Hz with a step size of 4.247 Hz.

The change in the filter's quality factor $Q$ is achievable by modifying the resistor $R_Q = 50 \cdot 10^3 \cdot Q$ (internal UAF42 resistance $R_{UAF} = 50$ kΩ).

The resistance values of the first and second potentiometers were then calculated for Bessel ($Q = 0.58$) and Butterworth ($Q = 0.707$) filters: $R_{V1} = 50 \cdot 10^3 \cdot 0.58 = 29$ kΩ, $R_{V2} = 50 \cdot 10^3 \cdot 0.707 \approx 35$ kΩ.

The sum of the two resistances above is approximately 64 kΩ, which means that for such a configuration, the equivalent resistance Q is equal to the following:

$$Q = \frac{R_Q}{R_{UAF}} = \frac{64\,\text{kΩ}}{50\,\text{kΩ}} = 1.28 \tag{6}$$

The calculated $Q$ is close to a Chebyshev filter with a 3 dB ripple, where the $Q = 1.3$. Therefore, for the calculated resistor values $R_{V1}$ and $R_{V2}$, three types of the most popular filters can be implemented.

### 3.3. Identification of Transfer Functions of the Active Filter Circuits

The transfer functions achievable in an analog filter system share a common denominator given by $s^2 + \frac{\omega}{Q}s + \omega^2$; see Section 2. The numerator varies depending on the filter

type: for a low-pass filter, it is $\omega^2$; for a high-pass filter, it is $s^2$; and for a band-stop filter, it is $s^2 + \omega^2$. The general form of the numerator for a band-pass filter is $\frac{\omega}{Q}s$. However, in the case of the ADF339 system, the numerator is additionally multiplied by the coefficient $Q$, thus simplifying it to the form $\omega s$. This simplification results in the gain of the band-pass filter not being equal to 1, as for other types of filters, but is determined by the formula $A = -20 \cdot \log_{10}\left(\frac{R_{V3}}{50k\Omega}\right) - 20 \cdot \log_{10}\left(\frac{1}{Q}\right)$ dB. For $f_c$ expressed in Hz, $\omega = f_c \cdot 2\pi$.

### 3.4. Applying a Phase Shift Compensation Circuit

A phase shifter based on the high-precision, low-noise OP37 operational amplifier powered by a symmetrical voltage of $\pm 9$ V is employed for the analog signal filtering. The phase shifter, shown in Figure 7, refers to a circuit designed to alter the phase relationship between the input and output analog signals.

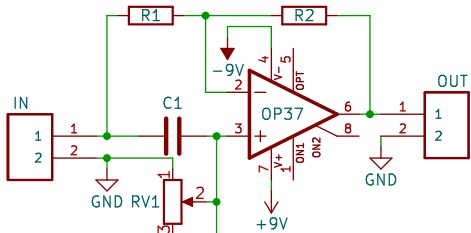

**Figure 7.** A phase shifter circuit used for compensation of phase of the output analog signal.

In the context of signal filtering in this work, the operational amplifier can be also configured to implement various filter designs, such as low-pass, high-pass, or band-pass filters, depending on the specific requirements of the application. The choice of a symmetrical power supply helps ensure a balanced and stable operation of the operational amplifier, thus contributing to the overall performance and accuracy of the phase shifter in filtering analog signals.

## 4. Programming the Digital Filtering and Data Acquisition FPGA-Based Circuit

In FPGA systems, there are structures called DSP48 slices designed for multiplication operations. The absence of such structures would lead to the consumption of multiple configurable logic block units for calculations. FPGAs also utilize RAM memory, thus allowing for storing computed results or signal states. In FPGA programming, this process is referred to as configuring, which considers the structural nature of the system. Configuring FPGA systems is done using hardware description languages, which are known for their high complexity. For the myRIO device, the NI provides the LabVIEW environment for configuring the FPGA system and programming the RT module, along with an extensive library designed for the myRIO-1900 device.

The *RTMain* schematic in Figure 8a includes elements of the transmission between the real-time (RT) system and the FPGA array, which occurs through a bidirectional DMA FIFO queue responsible for handling data transmission between these two subsystems.

The main tasks of the *RTMain* system include user HMI interface management, thus allowing for modification of the filter cutoff frequency and the $Q$ parameter, as well as displaying data on an oscilloscope. Additionally, the *RTMain* system is responsible for performing digital filtering using an IIR filter, whose coefficients are dynamically passed depending on the selected filter type through case structures.

In the context of digital filters, to obtain the equivalent of an analog filter, the bilinear transform is used, which calculates digital filter coefficients based on an analog filter. In the LabVIEW system, ready-made blocks are available that facilitate the generation of digital filter coefficients after entering parameters such as the center frequency $f_c$, the higher and

lower cutoff frequency $f_h$ and $f_l$, respectively ($\pm$), the quality factor $Q$, filter order, and sampling frequency:

$$f_{h,l} = f_c \cdot \left( \sqrt{1 + \frac{1}{4Q^2}} \pm \frac{1}{2Q} \right) \tag{7}$$

Alternatively, coefficients can be manually entered using a cluster. For the ADF339 band-pass filter, which has reduced gain in the pass-band, it is recommended to use the same method of entering filter coefficients, e.g., calculating them in MATLAB and entering them into the cluster.

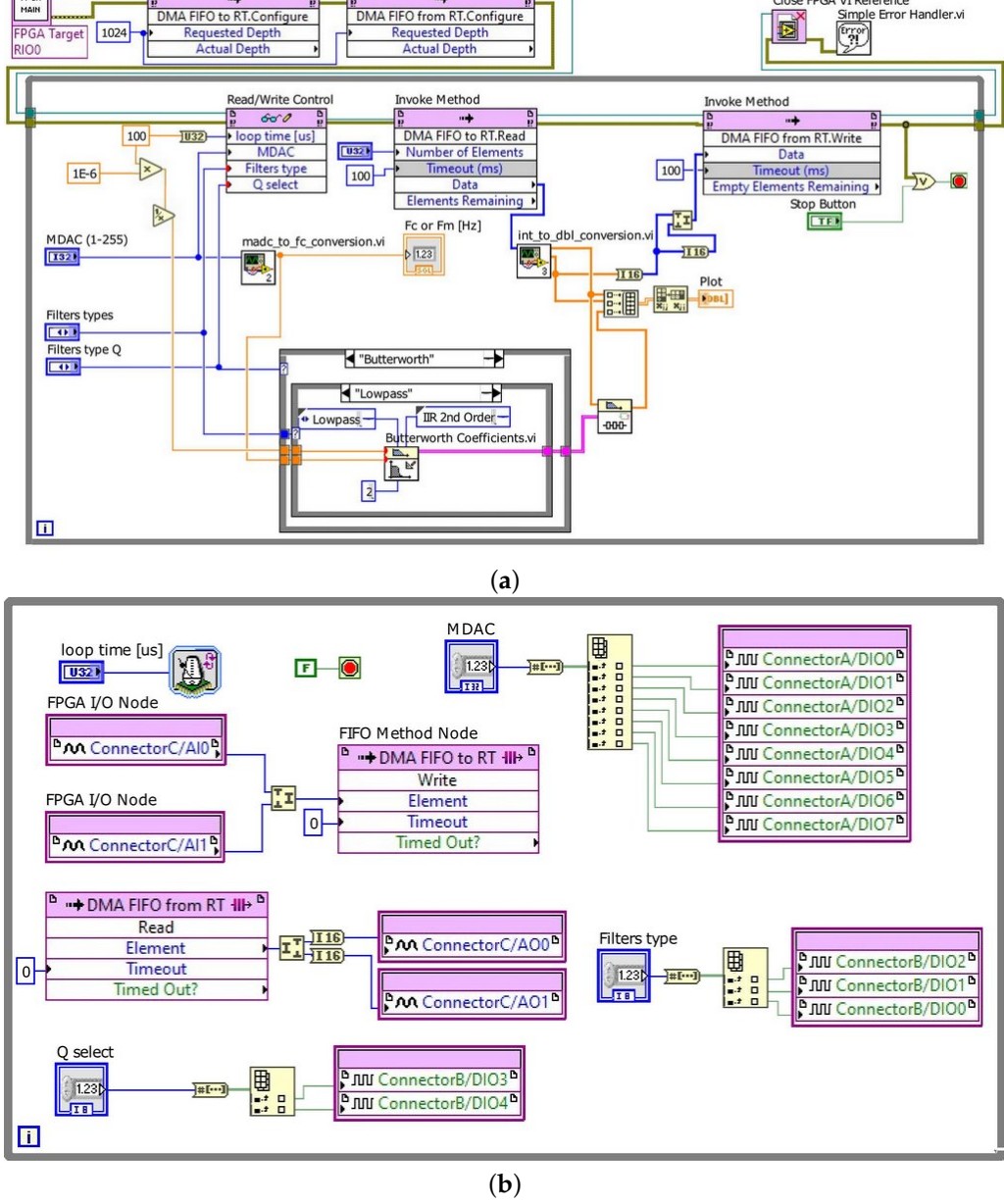

(**a**)

(**b**)

**Figure 8.** (**a**) A virtual instrument *RTMain* in LabVIEW establishing transmission between the RT system and the FPGA array. (**b**) A virtual instrument *FPGAMain* in LabVIEW receiving information about the selected cutoff frequency and then transferring it to an 8-bit data bus output, which controls MDAC units. Implementation of the numerical programming in diagrams of virtual instruments in LabVIEW allowing for receiving the analog signals from input ports of ADF339, realization of digital filtering, and sending the digital data to control settings of the analog signals.

The *FPGAMain* system in Figure 8b receives information about the selected cutoff frequency and transfers it to an 8-bit data bus output, which controls MDAC units. Similarly, information about the choice of filter type and the filter $Q$ factor is transferred through relay control. The *FPGA* system also samples the analog signal from two inputs: without filtering and after analog filtering.

By using a bidirectional DMA FIFO queue, data is transferred to the *RTMain* system, where it is displayed, subjected to digital filtering, and retrieved from the system. This also allows for outputting the signal to the DAC converter if needed.

Digital filtering and the DAC outputs of the myRIO-1900 device are an additional option, thus enabling a comparison between analog and digital filters. Through the use of digital control of an analog filter, the cutoff frequency can be adjusted to match the signal under investigation, thereby allowing for minimizing the phase shift. Another method to reduce the phase shift is the application of phase compensation techniques.

## 5. Results

### 5.1. Testing the Electronic Board ADF339

Signal comparisons involved measurements of three types: noisy, analog-filtered, and digitally filtered signals. To mitigate disturbances within the noisy signal, a consistent low-pass filtering approach was employed for both the analog and digital filters. This filtering process introduces a frequency-dependent phase shift in the input signal, with the magnitude of the shift varying based on factors such as the input signal frequency, filter type, and specific settings.

Low-pass filtering of a noisy signal using a constructed analog filter and its digital equivalent has been presented in Figure 9.

Generated from the universal signal generator, the specified signal $x(t)$ is a composition of a 500 Hz sinusoidal waveform with an amplitude of 10 Vpp and white noise with a standard deviation of 325 mV. Butterworth low-pass filtering at 1083 Hz was applied on the ADF339 board and on the FPGA myRIO-1900. The blue waveform represents the noisy input signal, the green waveform depicts the digitally filtered signal $\tilde{x}_d(t)$, and the red waveform illustrates the analog filtered signal $\tilde{x}_a(t)$.

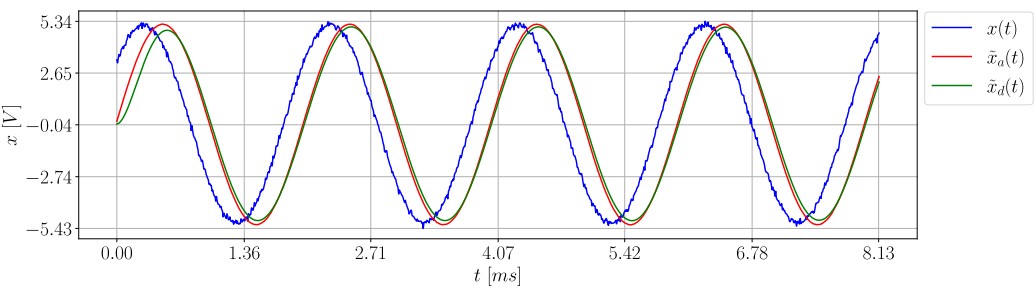

**Figure 9.** Butterworth low-pass analog (red) and digital (green) filtering of a real test noisy signal (blue) constructed as a sum of two signals in Ch. 1—sine 500 Hz, phase 0 deg., 10 Vpp—and in Ch. 2—noise, stdev 0.325 mV, mean 0 V.

Additionally, selected Bode plots presented in Figure 10, including the filter settings used to generate the waveform visible above in Figure 9, were plotted.

The phase shift for the analog filter was measured as 37.3 deg., while from the Bode plot obtained from computer simulation, it was 40.03 deg. For digital filtering, the measured phase shift from the time domain waveform was 40 deg., while for the simulation, it was 39.7 deg. All measurements were taken at a signal frequency of 500 Hz. In the case of the analog filter, the phase shift resulted from the inaccuracies of the passive elements used in the ADF339 circuit, which in the simulation were assumed to be ideal.

The band-pass filtering visible in Figure 11 was performed with a central frequency set at $f_{central} = 602.91$ Hz and utilizing Bessel topology. No phase shift was observed due to the closely matched filter central frequency and the input signal frequency. This

observation confirms the correct operation of the system in this configuration and the high accuracy of transmitting the signal to the output. The amplitude loss resulted form the amplitude frequency characteristics.

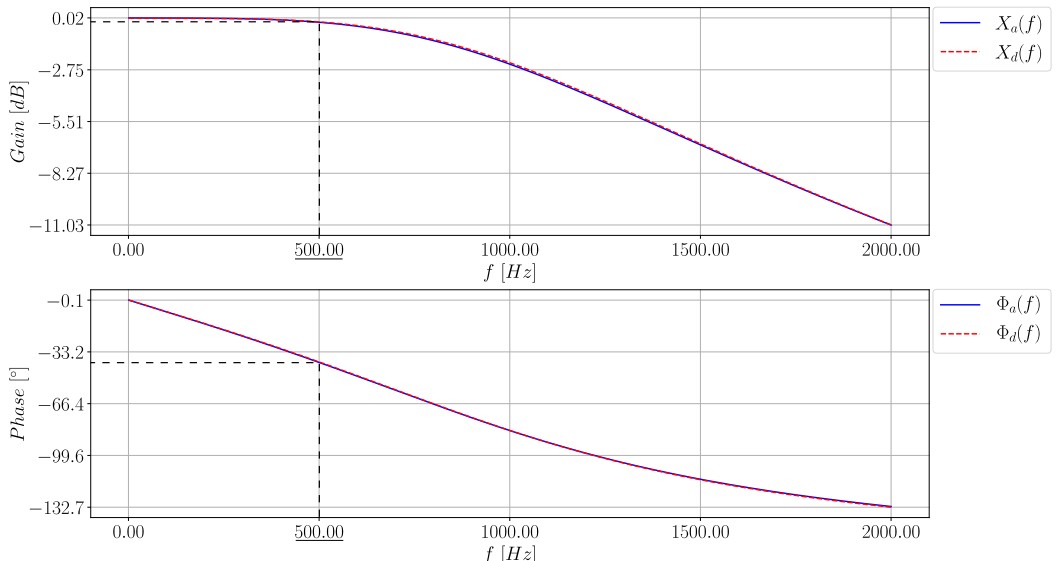

**Figure 10.** Bode plots informing about gain $X(f)$ and phase shift $\Phi(f)$ of the signals $\tilde{x}_a(t)$ and $\tilde{x}_d(t)$ illustrated in Figure 9.

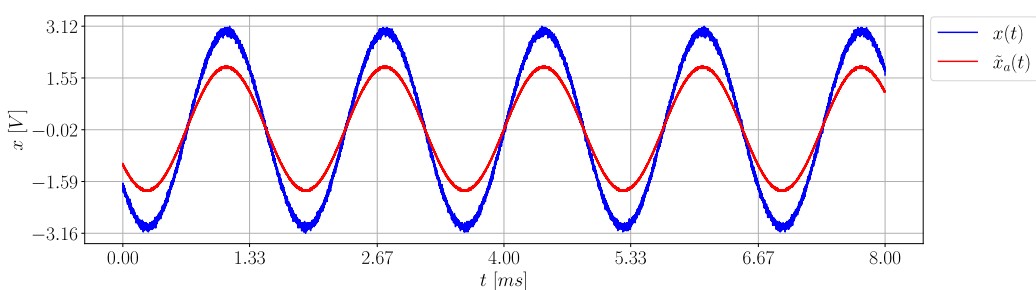

**Figure 11.** Band-pass analog (red) filtering of a real test noisy signal (blue) constructed as a sum of two signals in Ch. 1—sine 600 Hz, phase 0 deg., 6 Vpp—and in Ch. 2—noise, stdev 0.325 mV, mean 0 V.

In Figure 12, the effect of applying a low-pass filter with a phase shifter is demonstrated (see the electronic circuit in Figure 7). The obtained correct shift in the response $\tilde{x}_s(t)$ indicates a good effectiveness of the filtration within the frequency range of the investigated system's operation. The filtration was performed using a Butterworth topology with a cutoff frequency set at $f_c = 798.22$ Hz.

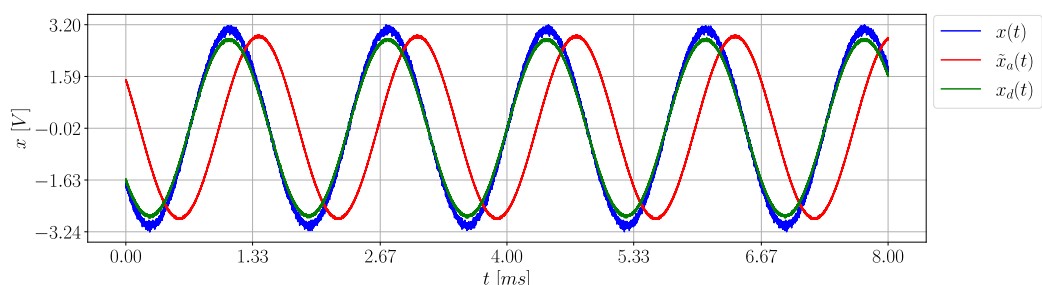

**Figure 12.** Low-pass analog (red) and phase-shifted analog (green) filtering of a real test noisy signal (blue) constructed as a sum of two signals in Ch. 1—sine 600 Hz, phase 0 deg., 6 Vpp—and in Ch. 2—noise, stdev 0.325 mV, mean 0 V.

### 5.2. Real Noisy Signals: Laboratory and Industrial Sensor Outputs

In this section, experimental measurements of the noisy signals from three real, original mechatronic system setups shown in Figure 13 have been conducted. The objective of the research is to evaluate the responses of the developed mixed mode universal analog ADF339 with FPGA-based digital control of parameters to real challenging signals, which exhibit oscillations affected by noise with various standard deviations and frequencies.

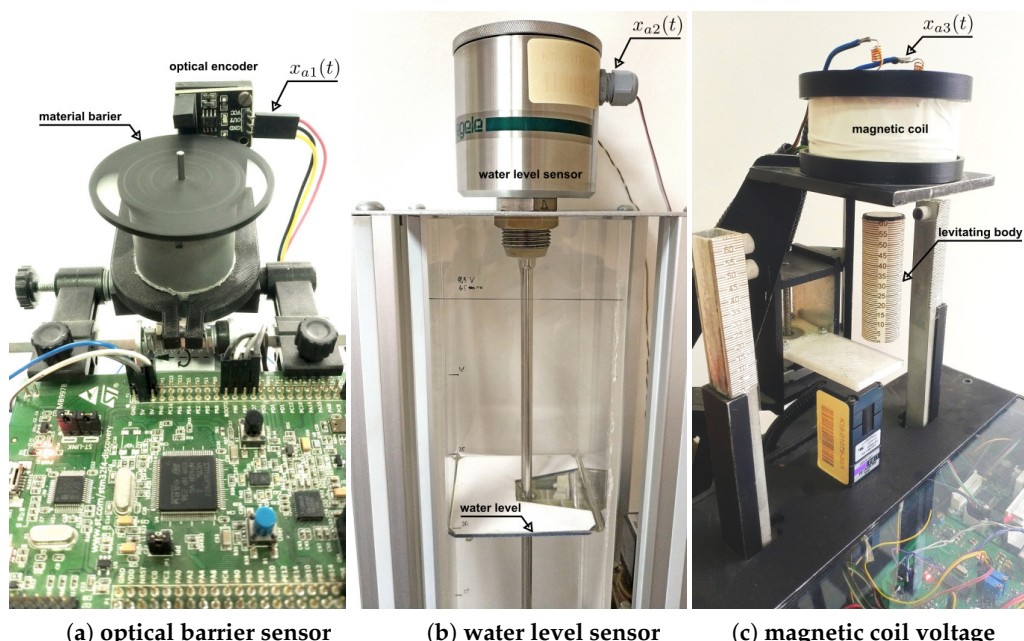

(**a**) **optical barrier sensor**     (**b**) **water level sensor**     (**c**) **magnetic coil voltage**

**Figure 13.** Laboratory and industrial mechatronic applications: Contactless measurement of rotational velocity of a DC motor with the use of infrared light barrier step-like voltage output (**a**). Measurement of water level [19] with the use of a two-electrode Negele sensor with continuous analog voltage output (**b**). Direct measurement of PWM-controlled analog voltage of an electromagnetic coil maintaining a levitating body at constant height (**c**).

The first signal came from the first laboratory test stand used for testing control methods utilizing a square waveform. An optical encoder was constructed to measure the rotational velocity of the motor shown in Figure 13a. These kinds of sensors, including light-based encoders, generate electrical pulses based on changes in light passing through apertures, thus making them commonly used for rotational speed measurement in applications such as motor control. The motor rotates the disc, thereby causing the alternating interruption of an infrared light beam emitted by a diode. This optical encoder served to measure the rotational speed of the motor reflected in the blue waveform $x_{a1}(t)$ shown in Figure 14. It can be seen that the elaborated board ADF339 filtered the noisy signal effectively.

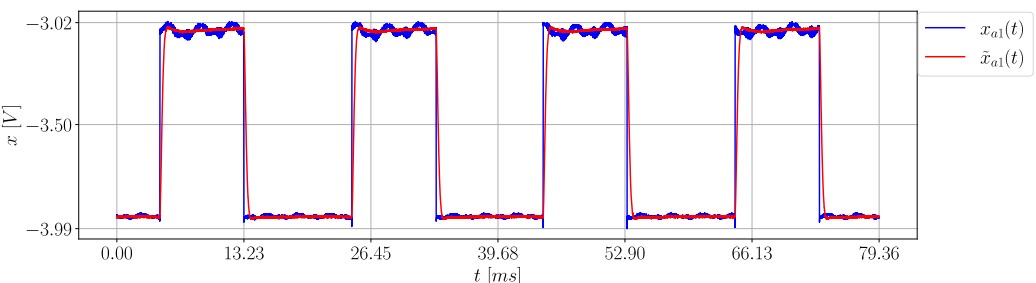

**Figure 14.** An optical encoder output signal $x_{a1}(t)$ and its filtered signal $\tilde{x}_{a1}(t)$ after using an analog low-pass Butterworth filter at $f_c = 1083$ Hz.

The second signal originated from another laboratory setup used to control the liquid level in a water-filled tank. The Negele sensor, installed as depicted in Figure 13b, has a power input and a current output loaded with an adjustable resistor. As a result, the system produced an output voltage signal $x_{a2}(t)$ with significant interference, stemming from the environmental conditions in which the laboratory system operates, as well as external devices such as a water pump or power supplies for power circuits. The filtration of this analog signal using the ADF339 board is illustrated in Figure 15.

Among the available configurations of analog filter circuits and frequencies in the electronic system, the optimal one was chosen. This involved a Bessel low-pass filter with a cutoff frequency of 42.47 Hz. However, due to numerous components of noise, amplitude variation in the input signal, and limitations of the adopted UAF42 integrated circuit configuration, the filtration of this signal yielded a modest gain.

The data acquired from the liquid level sensor were then digitally filtered with a sampling rate of 1 kHz for a low-pass filter with a cutoff frequency of 42 Hz, thus showing a similarly small benefit and response as that of the analog filter. However, by setting the cutoff frequency to $f_c = 1$ Hz (due to the slow dynamics of liquid level changes compared to the fast signals from the other two sensors used in the tests), a less noisy wave was obtained, as shown by signal $\tilde{x}_{d2}$ in Figure 15 below.

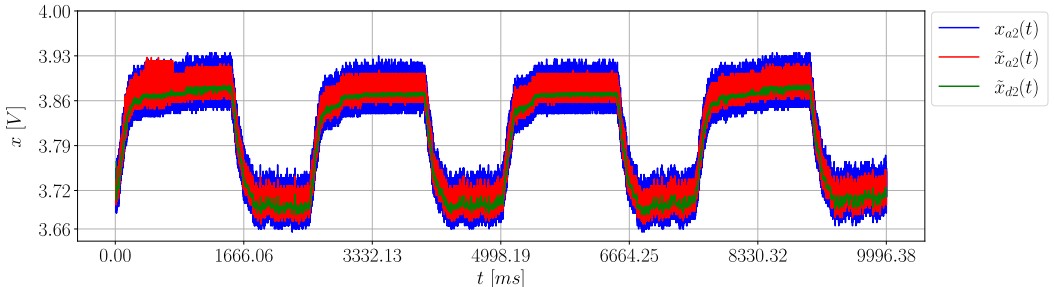

**Figure 15.** A water level sensor output signal $x_{a2}(t)$; its analog filtered signal $\tilde{x}_{a2}(t)$; and its digitally filtered signal $\tilde{x}_{d2}(t)$ after using an analog and digital low-pass Bessel filters at $f_c = 42.47$ Hz.

The third signal $x_{a3}(t)$ originated from the third laboratory setup used to investigate the phenomenon of magnetic levitation. In Figure 13c, a coil with a metal core served as the source of a magnetic field interacting with an object stably levitating in space. In the electronic control system, a magnetic field sensor measured the field's intensity and transmitted it to a PID controller. Through a power circuit, the PID controller adjusts the coil current by modulating the PWM signal duty cycle, which is measured at a high resistive load. It turns out, as commonly encountered in switching systems, that the electronic circuit generating the PWM signal exhibited high-frequency switching that interfered with the regulation. This impact affected the quality of stabilizing the levitating object in a dynamically balanced position. The effect of filtering this signal using the tested ADF339 is shown in Figure 16.

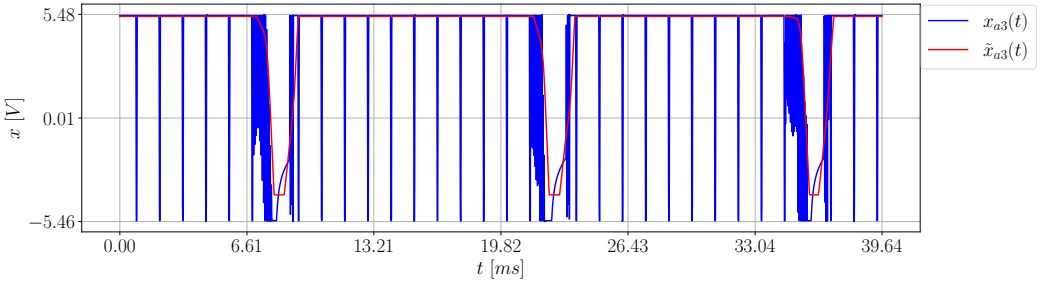

**Figure 16.** A magnetic coil control signal $x_{a3}(t)$ and its filtered signal $\tilde{x}_{a3}(t)$ after using an analog low-pass Bessel filter at $f_c = 849.17$ Hz.

It can be observed that the effectiveness of the performed filtration was high, thus resulting in an improvement in the stabilization of the levitating mass.

### 5.3. Performance Metrics

Performance metrics using the pSNR (power signal-to-noise ratio) for filter responses to an input signal was employed to assess the effectiveness of a filter in isolating the desired signal from unwanted noise, e.g., see [3].

During the evaluation, a quantitative measure of how well a filter preserves the desired signal while attenuating noise was found. Power SNR metrics provided insights into the overall quality of the filtered signal, thereby helping us to identify potential issues such as distortion or loss of signal fidelity. It was calculated by means of the formula:

$$\text{pSNR} = 10 \cdot \log_{10}\left(\frac{\text{signal}}{\text{noise}}\right) \tag{8}$$

The ratio was computed in the measured time interval of observation based on the provided 'signal' and 'noise' waveforms.

For example, using Equation (8) to measure the DC motor system, the calculations applied the square of the signal's amplitude, while in the case of noise power, it was calculated as the sum of the squares of the frequency spectrum components, thus excluding the bars responsible for the odd harmonics of the square wave signal. Depending on the number of assumed harmonics of the square wave signal, the pSNR varied, but for approximately 10 harmonics used, the value is presented in the first column of Table 1.

**Table 1.** Power SNR of the analog filter responses illustrated in Figures 14–16 (ratios of signals before filtration are written in brackets).

| Optical Barrier Sensor | Water Level Sensor | Coil Voltage Sensor |
|:---:|:---:|:---:|
| 14.57 (11.68) | 13.23, 15.25$^{(\text{digital})}$ (11.82) | 20.66 (2.60) |

The filtration process effectively reduced the impact of noise in the signals $x_{a1}(t)$ and $x_{a3}(t)$, thus leading to a much higher pSNR. In practical terms, the higher pSNR values suggest that the observed signals are more distinguishable from the background noise, and this improvement will be beneficial in the presented applications, where a clear and reliable signal is desired. Exceptionally, the second case and lower pSNR values of the signal $x_{a2}(t)$ need the digital approach, in which $15.25^{(\text{digital})}$ has been achieved; see the $\tilde{x}_{d2}(t)$ waveform in Figure 15. The results of this test indicate a limitation in the minimum cutoff frequency available on the examined board.

The higher pSNR values observed for the filtered signal from the optical encoder and sourcing the magnetic coil indicates better performance of the ADF339 board in filtering square signals, as it signifies a stronger and clearer signal relative to the noise level.

### 6. Conclusions

This study introduced and also deeply explored two cooperating electronic subsystems of the ADF339 analog and digital filter circuit that underwent testing under real laboratory conditions. Filtering was performed, thereby demonstrating its effectiveness on a series of actual analog measurement signals recorded in various measurement channels. Utilizing a fast measurement system based on an FPGA array, raw and filtered voltage signals from measurement lines operating in a highly disturbed environment were recorded. In the case of one complex and heavily distorted signal, the effectiveness of the investigated unit was low; however, the remaining signals were smoothed very well. Further development of the presented ADF339 board's prototype is anticipated, including its permanent integration into several measurement channels and enhancement of its filtering capabilities.

**Author Contributions:** Conceptualization, C.P. and P.O.; methodology, C.P.; software, C.P.; validation, C.P.; formal analysis, C.P. and P.O.; investigation, C.P. and P.O.; resources, P.O.; data curation, C.P. and P.O.; writing—original draft preparation, P.O.; writing—review and editing, C.P. and P.O.; visualization, C.P. and P.O.; supervision, P.O.; project administration, C.P.; funding acquisition, P.O. All authors have read and agreed to the published version of the manuscript.

**Funding:** This research received no external funding.

**Institutional Review Board Statement:** Not applicable.

**Informed Consent Statement:** Not applicable.

**Data Availability Statement:** The data are available upon reasonable request from the authors.

**Conflicts of Interest:** The authors declare no conflicts of interest.

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
