# Peer review of "An Experimental Electronic Board ADF339 for Analog and FPGA-Based Digital Filtration of Measurement Signals"

_electronics, doi:10.3390/electronics13040805_

Round 1
Reviewer 1 Report
Comments and Suggestions for Authors
This paper introduces an electronic system, which is used to filter analog measurement signals of low frequencies. The system is used for real laboratory setups. Besides, the authors also propose to use a signal-to-noise ratio to indicate the quality of the application. I would like to recommend accepting this paper with minor revisions on format and style. For instance, it is not necessary to use [] for a unit.
Author Response
Dear Reviewer,
Thank you for your efforts in reviewing this manuscript and for your valuable remarks. We have addressed all the corrections we noticed, enhancing the overall presentation of our research results (our responses are marked in green).
Sincerely,
Authors
Comments in the Review 1
This paper introduces an electronic system, which is used to filter analog measurement signals of low frequencies. The system is used for real laboratory setups. Besides, the authors also propose to use a signal-to-noise ratio to indicate the quality of the application. I would like to recommend accepting this paper with minor revisions on format and style. For instance, it is not necessary to use [] for a unit.
Corrected. Thank you.
Reviewer 2 Report
Comments and Suggestions for Authors
The work proposes a new programmable electronic system designed for filtering analog measurement signals of low frequencies, which operates in a mixed mode using a digital controller implemented on a FPGA module. The authors tested this system on various signals occurring in real laboratory setups. A series of real responses from three different laboratory systems and a measurement system using LabVIEW virtual instruments are presented.
The paper is overall well written, the introduction with literature survey is comprehensive, references are adequate and up-to-date .
Some aspects of the paper, mainly regarding filters, are rather elementary. The authors should highlight more clearly their original contributions.
Please mainly revise the following issues:
1) "maximally flat FIR differentiator" ? In what sense the differentiator is maximally flat? As is well known, the transfer function of the analog differentiator is simply H(s) = s = j*w ; please give more explanations.
2) line 64: the acronym "LLCL" is not explained
line 79: filter Q -> quality factor Q
line 153: The order of the circuit is determined by the maximum degree of s present in the network function -> The order of the circuit is determined by the degree of the transfer function's denominator
line 163: Poles and zeros generally exist as complex entities -> Poles and zeros generally exist as complex-conjugated pairs
3) The paragraph between lines 121-124 is repeated identically at lines 171-174; please express it otherwise.
Caption of figure 1 is too long; abbreviations should be explained in the main text instead.
In section 3, measurement units should be given directly, not between brackets, for instance 64 kΩ, not 64 [kΩ] etc.
4) Please explain the use of the two digital damping systems shown in figure 3, their role is not very clear . Since they are "almost identical", why both of them were given and which one is actually used, what is the functional difference between them, is one better than the other?
The two schematics in figure 3 should be better placed one above the other since they are independent, otherwise they are difficult to be viewed separately.
5) In subsection 3.1: "To eliminate switching using a switch, an output selector was implemented using three relays" - please explain why the output selector with relays is superior to the usual switching.
6) line 248: "where in Figure 3: RF1 = R10 + R12, RF2 = R9 + R11" ; in figure 3, these resistances cannot be found, they are not marked clearly, please verify and correct.
line 338: analog filtering process (?) -> analog filtered signal
Regarding the signals shown in figure 9, please explain why the analog filtered signal has a larger phase shift than the digitally filtered signal, compared to the original noisy signal.
Maybe it would be useful to represent comparatively the phase characteristics of the two filters (analog and digital).
7) Regarding the band-pass filtering shown in figure 10, a band-pass filter has two cutoff frequencies (or lateral frequencies, inferior and superior); the signal is not phase shifted when its frequency is equal to the filter central (peak) frequency (where usually the phase shift is zero), not is cutoff frequency ; please check and correct.
line 343: high accuracy of faithfully transmitting frequency of the signal to the output -> high accuracy of transmitting the signal to the output
line 364: "reflected in the blue time history" (?) - please reformulate ("reflected in the blue waveform" ? )
line 376: frequency of 42.47 -> frequency of 42.47 Hz
8) GRAMMAR: Grammar is overall good, just a few corrections should be made, as suggested below:
line 34: Another FPGA-based digital filters were designed -> Another FPGA-based digital filter was designed / Other FPGA-based digital filters were designed
line 63: power electronics-based devices -> power electronics devices
line 89: The`motion sensor signal developed -> The signal from the motion sensor developed
line 91: UAF filter prevents noise -> UAF filter eliminates / rejects noise
line 92: quality -> quality factor
line 101: electrocardio signal -> ECG signal
Title of subsection 1.5: "Summary of the Overview" (?)
line 111: The selected and current applications -> Some applications / A few applications
line 112: filters created using FPGA -> filters implemented on FPGA
line 115: collaboration between -> joint operation of (in general, "collaboration" refers to human actions)
line 117: collaboration -> joint operation ; enhances the tuning parameters -> enhances the tuning of the parameters
line 121: "the desirability lies in circuits" - please reformulate
line 125: A low-pass frequency filter -> A low-pass filter
electronically adjustable approximation characteristics -> electronically adjustable characteristics / frequency response
line 134: "angular frequency" - please check and correct (cutoff / central frequency etc.)
lines 136, 137: Antialiasing -> Anti-aliasing
making them necessary as analog filters (?) - verify and reformulate
line 141: "The enumerated actual fields .... " - reformulate the entire sentence
line 171: the desirability lies in circuits characterized by ... -> circuits characterized by ... are preferred
line 175: will be used and precised (?)
line 182: The block diagram ... simplifies the communication -> The block diagram ... shows in a simplified form the communication
lines 228, 231: model programmable -> model of programmable
line 398: During the quantitative evaluation a quantitative measure -> During the evaluation, a quantitative measure
line 408: The filtration processes has effectively reduced -> The filtration process has effectively reduced
Comments on the Quality of English LanguageThe language, grammar and style are overall good, however some corrections should be made, as notified to authors.
Author Response
Dear Reviewer,
Thank you for your efforts in reviewing this manuscript and for your valuable remarks. In a separate PDF file we have addressed all the corrections we noticed, enhancing the overall presentation of our research results (our responses are marked in green).
Sincerely,
Authors

Reviewer 3 Report
Comments and Suggestions for Authors
This paper presents a programmable electronic system, tailored to filter analog signals at low frequencies. The selection about the kind of filtering (approximation and filter category), as well as its frequency parameters, is carried out digitally. A mix of analog and digital filters implemented in FPGA is experimented, yielding a mixed mode system, aiming to enhance the tuning parameters, frequency response, phase shift, and the range of applications. A literature review was undertaken to summarize the state-of-the-art publications and how the proposed work fits into the issue under analysis. A basic mathematical analysis was performed to review the main equations that are used when dealing with analog filtering. The design of the proposed system is then presented in detail, by bringing out the options that were made in its conception. The proposed approach was tested using a laboratorial setup, encompassing a set of given scenarios. The results were satisfactory when dealing with the noise suppression desired for those situations. The quality of the written English is very good.
However, there are some remarks regarding the content of the paper, to clarify some aspects, and to fix some small issues, as follows:
Mark 71 – “AO” should be replaced with “OA”.
Mark 192 – It should be explained what the purpose for the potentiometer RV3 is, and why it was selected to be 49.9k.
Mark 233 – The reason that led to the selection of relays, instead of another kind of switches, should be emphasized. How is addressed the electromechanical noise, caused by the relays?
Mark 334 – x(t), shown in Figure 9, is 5 V peak, not peak to peak (Vpp).
For the sake of completeness, the results shown in Figure 9 lack some more explanations, such as the value of the delay (or phase shift), introduced by each of the filtering methods, as well as the slight offset noticed in the output signal obtained by analog means. Moreover, it would also be valuable to have a perspective of the Bode plots (both amplitude and phase) of each of the filter approximations employed. As a matter of fact, no Bode plots are presented throughout the manuscript. Only signals in the time domain are shown, which can be misleading.
In general, a comparison should be provided about the desired filtering transfer functions, and those that were obtained by using the proposed system, by overlapping with one another.
Mark 376 – The cutoff frequency units are missing.
Mark 401 – Preferably, the SNR is expressed as a ratio of powers rather than amplitudes.
The discussion about the data shown in Table 1 needs more substance, as it is very simplistic. The aspects that were pointed out should be treated in more depth. For instance, in mark 412 - “The second case and lower SNR values of the signal xa2(t) need different approach but the results show some restriction in use of the investigated board.” – What kind of approach would eventually get it to perform better?
Author Response

(The authors gave the same response as above.)

Round 2
Reviewer 3 Report
Comments and Suggestions for Authors
This paper presents a programmable electronic system, tailored to filter analog signals at low frequencies. The selection about the kind of filtering (approximation and filter category), as well as its frequency parameters, is carried out digitally. A mix of analog and digital filters implemented in FPGA is experimented, yielding a mixed mode system, aiming to enhance the tuning parameters, frequency response, phase shift, and the range of applications. A literature review was undertaken to summarize the state-of-the-art publications and how the proposed work fits into the issue under analysis. A basic mathematical analysis was performed to review the main equations that are used when dealing with analog filtering. The design of the proposed system is then presented in detail, by bringing out the options that were made in its conception. The proposed approach was tested using a laboratorial setup, encompassing a set of given scenarios. The results were satisfactory when dealing with the noise suppression desired for those situations. The quality of the written English is very good.
After analyzing this re-submission, and as far as I am concerned, the comments on the previous submission have been generally addressed. The additional content that has been elaborated helped the paper to get more robust and enriched.
As such, I foresee that the actions to be taken by the authors should be to comply with any remarks subsisting from the remaining reviewers. From my point of view, the paper can be published as it is and, once again, provided that the conditions imposed by the remaining reviewers are fully met.